# Genome-wide association study of self-reported walking pace suggests beneficial effects of brisk walking on health and survival

Iain R. Timmins[1], Francesco Zaccardi [2], Christopher P. Nelson[3,4], Paul Franks [5], Thomas Yates[2,4] & Frank Dudbridge [1✉]

Walking is a simple form of exercise, widely promoted for its health benefits. Self-reported walking pace has been associated with a range of cardiorespiratory and cancer outcomes, and is a strong predictor of mortality. Here we perform a genome-wide association study of self-reported walking pace in 450,967 European ancestry UK Biobank participants. We identify 70 independent associated loci ($P < 5 \times 10^{-8}$), 11 of which are novel. We estimate the SNP-based heritability as 13.2% (s.e. = 0.21%), reducing to 8.9% (s.e. = 0.17%) with adjustment for body mass index. Significant genetic correlations are observed with cardiometabolic, respiratory and psychiatric traits, educational attainment and all-cause mortality. Mendelian randomization analyses suggest a potential causal link of increasing walking pace with a lower cardiometabolic risk profile. Given its low heritability and simple measurement, these findings suggest that self-reported walking pace is a pragmatic target for interventions aiming for general benefits on health.

---

[1] Department of Health Sciences, University of Leicester, Leicester, UK. [2] Diabetes Research Centre, University of Leicester, Leicester, UK. [3] Department of Cardiovascular Sciences, University of Leicester, Leicester, UK. [4] NIHR Leicester Biomedical Research Centre, University Hospitals of Leicester NHS Trust & University of Leicester, Leicester, UK. [5] Department of Clinical Sciences, Lund University, Lund, Sweden. ✉email: frank.dudbridge@leicester.ac.uk

Walking is a simple and convenient form of exercise that is widely promoted for its benefit to physical fitness and overall health[1]. The public health recommendations for walking focus particularly on increasing the time spent walking and the number of steps walked, with walking at a faster pace receiving less emphasis[2].

However, recent studies have observed a brisk habitual walking pace, self-reported through questionnaire or verbal interview, to be associated with reduced risk of a range of cardiorespiratory and cancer outcomes[2,3]. Most notably, self-reported habitual walking pace has been identified as one of the strongest predictors of all-cause mortality[4], even when adjusting for the effects of established risk factors such as body mass index (BMI)[5] and other lifestyle behaviours including smoking[6].

Despite the strong associations of self-reported walking pace with health and survival, it is unclear whether these associations arise from common biological processes, including genetic predisposition, nor whether there are causal effects of walking pace on health outcomes. These questions can be addressed with knowledge of the genetics of walking pace. To date, studies examining the genetic component of walking pace have analysed objectively measured gait speed, where speed is assessed by timing participants to walk a distance of up to 8 m. These studies focussed on older adults, giving insight into the biological mechanisms underlying age-related diseases and physical mobility[7,8]. Genome-wide significant markers of objectively measured gait speed were not identified in these studies, which had a maximum sample size of 31,479.

To examine the genetic component of self-reported walking pace, we performed a genome-wide association study (GWAS) in UK Biobank, a prospective study of approximately 450,000 adults of European descent, in addition to approximately 50,000 participants of other ethnicities, aged between 40 and 69 years at baseline[9]. Participants self-reported their walking pace as "slow", "steady/average" or "brisk". We aimed to identify associated genetic variants and their possible function, quantify the genetic correlation of walking pace with other complex traits, and assess the potential of self-reported walking pace as a modifiable health-related exposure. Through these analyses we identify 70 genetic loci for self-reported walking pace and show that this trait shares its genetic architecture with other cardiometabolic risk factors, including educational attainment and cognitive outcomes. Using Mendelian randomisation (MR) we find evidence in favour of causal relationships between self-reported walking pace and several traits associated with mortality. This suggests that self-reported walking pace may indeed be a logical target of health interventions.

## Results

**GWAS of self-reported walking pace identifies 70 associated loci.** We performed a GWAS of self-reported walking pace in 450,967 individuals of European ancestry from UK Biobank (full details in Methods). The phenotype was coded 0, 1 and 2 for self-reported slow, steady/average and brisk walking pace, and the characteristics of participants across these categories are summarised in Supplementary Data 1. We used a linear mixed model with covariates for age, sex, genotyping array and 20 principal components of ancestry implemented in BOLT-LMM v2.3.3[10]. After quality control 10,061,374 imputed variants were analysed (Fig. 1). We identified 144 independent significant SNPs across 70 genomic loci (Table 1), indexed by 75 lead SNPs (Supplementary Data 2).

We estimated an inflation in the test statistics ($\lambda_{GC} = 1.597$, mean $\chi^2 = 1.767$) but, similarly to other phenotypes analysed in UK Biobank[11], the LD score intercept of 1.058 (s.e. = 0.0120)

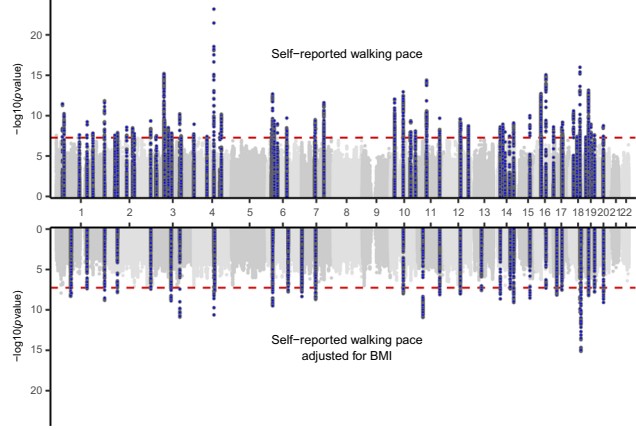

**Fig. 1 Miami plot of self-reported walking pace GWAS results with and without adjustment for BMI.** The x-axis is ordered by chromosome and base position. On the y-axis the −log10(P-value) is shown, where P-values are from a Wald test in the BOLT-LMM mixed model test of association ($N = 450,967$ individuals). A genome-wide significance threshold of $P < 5 \times 10^{-8}$ is indicated by the red dotted line.

suggests that the inflation is largely due to polygenic signal and the large sample size rather than population substructure.

As there is a clear negative association between BMI and self-reported walking pace (Supplementary Data 1), we were concerned that the results may simply reflect genetic associations with BMI, which have been extensively described[12]. We therefore performed a sensitivity analysis by including BMI as a covariate in the model (Fig. 1). Of the 70 associated loci only 15 retained genome-wide significance following adjustment for BMI, whilst 45 loci in total maintained a suggestive significance level ($P < 10^{-5}$). In addition, using LD score regression[13] we observed a strong genetic correlation between self-reported walking pace with and without adjustment for BMI ($r_g = 0.83$, s.e. = 0.0073), suggesting that much of the genetic component of walking pace is independent of BMI.

**Post-GWAS annotation, gene-based analysis and tissue-enrichment analyses.** A detailed annotation catalogue of candidate SNPs in the associated genomic loci is presented in Supplementary Data 3. Of the 70 independently associated genomic loci, 59 have previously reported suggestive associations for other traits and diseases (Supplementary Data 4). The strongest overlaps with the self-reported walking pace include 28 shared loci with BMI, 20 loci associated with educational attainment and 13 loci associated with hand grip strength.

Using positional mapping and expression quantitative trait loci (eQTL) mapping, we identified a total of 535 genes associated with genome-wide significant SNPs (Supplementary Data 5). We also performed a genome-wide gene-based association study (GWGAS) that identified 255 genes associated with self-reported walking pace (Supplementary Data 6), of which 152 were implicated through positional or eQTL mapping.

The strongest self-reported walking pace signals were identified within *SLC39A8* on chromosome 4, which has previously been associated with metabolic traits[14], *FTO* on chromosome 16, strongly associated with fat mass and obesity[15], and *TCF4* on chromosome 18, linked to neurocognitive traits and psychiatric disease[16]. Of these, *SLC39A8* and *TCF4* remained genome-wide significant after adjustment for BMI, while the association of *FTO* was attenuated as expected but remained nominally associated (Supplementary Data 2).

**Table 1 Seventy independent loci associated with self-reported walking pace at genome-wide significance ($P < 5 \times 10^{-8}$).**

| SNP | Chr | Position | Implicated gene | Function | EA/NEA | MAF | Self-reported walking pace | | Self-reported walking pace adjusted for BMI | | Novel |
|---|---|---|---|---|---|---|---|---|---|---|---|
| | | | | | | | Beta | P-value | Beta | P-value | |
| rs12739999 | 1 | 32,207,990 | ADGRB2 | Intronic | G/A | 0.17 | 0.016 | $3.44 \times 10^{-12}$ | 0.0084 | $1.58 \times 10^{-7}$ | No |
| rs113825410 | 1 | 40,057,543 | PABPC4 | Intergenic | A/G | 0.22 | 0.0098 | $5.52 \times 10^{-11}$ | 0.0068 | $1.87 \times 10^{-6}$ | No |
| rs699785 | 1 | 117,200,750 | IGSF3 | Intronic | G/A | 0.24 | -0.0082 | $2.29 \times 10^{-8}$ | -0.0066 | $2.94 \times 10^{-6}$ | Yes |
| rs11548200 | 1 | 155,028,522 | ADAM15 | Intronic | G/A | 0.47 | 0.0071 | $2.53 \times 10^{-10}$ | 0.0055 | $5.31 \times 10^{-6}$ | Yes |
| rs11264302 | 1 | 156,290,656 | CCT3 | Exonic | T/C | 0.07 | 0.0156 | $5.51 \times 10^{-10}$ | 0.0111 | $4.67 \times 10^{-6}$ | No |
| rs10797999 | 1 | 185,137,628 | SWT1 | Intronic | C/T | 0.41 | -0.0072 | $1.42 \times 10^{-8}$ | -0.0054 | $8.84 \times 10^{-6}$ | No |
| rs12127073 | 1 | 243,614,705 | SDCCAG8 | Intronic | C/G | 0.11 | -0.0141 | $1.40 \times 10^{-12}$ | -0.0115 | $1.60 \times 10^{-9}$ | No |
| rs1531133 | 2 | 46,843,631 | PIGF | Intronic | A/G | 0.42 | -0.0071 | $2.46 \times 10^{-8}$ | -0.0051 | $2.79 \times 10^{-5}$ | No |
| rs30005495 | 2 | 60,157,097 | BCL11A | ncRNA | T/G | 0.42 | 0.0072 | $1.16 \times 10^{-8}$ | 0.0054 | $1.07 \times 10^{-5}$ | No |
| rs55680124 | 2 | 105,984,624 | FHL2 | ncRNA | C/T | 0.16 | 0.0102 | $2.57 \times 10^{-9}$ | 0.0072 | $1.09 \times 10^{-5}$ | No |
| rs17698630 | 2 | 135,691,725 | CCNT2 | Intronic | A/G | 0.18 | -0.0097 | $3.12 \times 10^{-9}$ | -0.0074 | $1.92 \times 10^{-6}$ | No |
| rs5026760 | 2 | 144,137,353 | ARHGAP15 | ncRNA | A/G | 0.17 | -0.0096 | $1.64 \times 10^{-8}$ | -0.0083 | $3.02 \times 10^{-7}$ | No |
| rs62054079 | 2 | 226,486,752 | NYAP2 | Intronic | C/T | 0.32 | -0.0084 | $4.30 \times 10^{-10}$ | -0.0069 | $6.68 \times 10^{-8}$ | No |
| rs62246314 | 3 | 9,504,099 | SETD5 | Intronic | G/A | 0.10 | 0.0114 | $4.10 \times 10^{-8}$ | 0.0073 | $2.52 \times 10^{-4}$ | No |
| rs2920503 | 3 | 12,324,230 | PPARG | UTR5 | C/T | 0.29 | -0.0078 | $2.91 \times 10^{-8}$ | -0.0051 | $1.49 \times 10^{-4}$ | No |
| rs2280406 | 3 | 49,941,436 | MST1R | Intronic | G/A | 0.49 | 0.0101 | $6.10 \times 10^{-16}$ | 0.0048 | $6.81 \times 10^{-5}$ | No |
| rs6798941 | 3 | 52,893,465 | STIMATE | Intronic | C/T | 0.30 | 0.0086 | $3.32 \times 10^{-10}$ | 0.0051 | $8.09 \times 10^{-5}$ | No |
| rs830627 | 3 | 71,675,270 | FOXP1 | Intergenic | G/A | 0.42 | -0.0076 | $2.93 \times 10^{-9}$ | -0.0059 | $1.05 \times 10^{-6}$ | No |
| rs114547690 | 3 | 88,100,210 | CGGBP1 | Intergenic | A/G | 0.12 | 0.0110 | $1.38 \times 10^{-8}$ | 0.0063 | $5.53 \times 10^{-4}$ | No |
| rs6763292 | 3 | 129,044,705 | H1FX | ncRNA | A/G | 0.22 | -0.0098 | $5.81 \times 10^{-11}$ | -0.0098 | $1.19 \times 10^{-11}$ | No |
| rs9844666 | 3 | 135,974,216 | PCCB | UTR5 | G/A | 0.24 | 0.0081 | $2.97 \times 10^{-8}$ | 0.0047 | $7.76 \times 10^{-4}$ | No |
| rs798750 | 4 | 1,717,171 | TMEM129 | Intergenic | C/T | 0.38 | 0.0072 | $1.82 \times 10^{-8}$ | 0.0065 | $8.84 \times 10^{-8}$ | No |
| rs362307 | 4 | 3,241,845 | HTT | UTR3 | T/C | 0.08 | 0.0146 | $1.11 \times 10^{-9}$ | 0.0094 | $3.75 \times 10^{-5}$ | No |
| rs72636700 | 4 | 68,019,509 | CENPC | Intergenic | C/T | 0.17 | 0.0092 | $3.57 \times 10^{-8}$ | 0.0079 | $7.92 \times 10^{-7}$ | No |
| rs13107325 | 4 | 103,188,709 | SLC39A8 | Exonic | C/T | 0.07 | 0.0240 | $6.31 \times 10^{-24}$ | 0.0153 | $2.22 \times 10^{-7}$ | No |
| rs115202226 | 4 | 133,802,757 | PCDH10 | ncRNA | A/G | 0.01 | -0.0489 | $1.62 \times 10^{-8}$ | -0.0399 | $1.48 \times 10^{-6}$ | Yes |
| rs57800857 | 4 | 140,863,365 | MAML3 | Intronic | A/C | 0.37 | -0.0085 | $6.38 \times 10^{-11}$ | -0.0056 | $6.02 \times 10^{-6}$ | No |
| rs4134943 | 6 | 20,483,407 | E2F3 | Intronic | C/T | 0.20 | -0.0091 | $4.99 \times 10^{-9}$ | -0.0079 | $1.62 \times 10^{-7}$ | No |
| rs9366651 | 6 | 26,336,696 | BTN3A2 | Intergenic | G/T | 0.49 | -0.0093 | $2.00 \times 10^{-13}$ | -0.0076 | $3.04 \times 10^{-10}$ | No |
| rs1061801 | 6 | 33,282,338 | ZBTB22 | UTR3 | G/A | 0.19 | 0.0087 | $4.61 \times 10^{-8}$ | 0.0052 | $6.10 \times 10^{-4}$ | No |
| rs205262 | 6 | 34,563,164 | C6orf106 | Intronic | A/G | 0.27 | 0.0088 | $4.11 \times 10^{-10}$ | 0.0035 | $9.20 \times 10^{-3}$ | Yes |
| rs4715208 | 6 | 50,829,471 | TFAP2B | Intergenic | A/G | 0.25 | 0.0089 | $1.04 \times 10^{-9}$ | 0.0033 | $1.94 \times 10^{-2}$ | No |
| rs11152989 | 6 | 96,936,061 | UFL1 | ncRNA | C/T | 0.31 | 0.0080 | $3.13 \times 10^{-9}$ | 0.0062 | $1.69 \times 10^{-10}$ | Yes |
| rs4839898 | 6 | 97,546,759 | KLHL32 | Intronic | G/A | 0.11 | -0.0130 | $1.95 \times 10^{-10}$ | -0.0106 | $5.58 \times 10^{-8}$ | No |
| rs7804774 | 7 | 66,903,028 | TYW1 | Intergenic | A/G | 0.19 | -0.0089 | $3.24 \times 10^{-8}$ | -0.0081 | $1.49 \times 10^{-7}$ | No |
| rs10452738 | 7 | 69,453,714 | AUTS2 | Intronic | A/G | 0.32 | 0.0085 | $2.99 \times 10^{-10}$ | 0.0063 | $8.23 \times 10^{-7}$ | No |
| rs7795394 | 7 | 113,560,607 | PPP1R3A | Intronic | T/A | 0.38 | -0.0096 | $2.43 \times 10^{-12}$ | -0.0059 | $1.57 \times 10^{-6}$ | No |
| rs1243184 | 10 | 21,931,937 | MLLT10 | Intronic | T/C | 0.32 | 0.0096 | $8.81 \times 10^{-13}$ | 0.0059 | $5.29 \times 10^{-6}$ | No |
| rs7924036 | 10 | 65,191,645 | JMJD1C | Intronic | G/T | 0.50 | -0.0093 | $1.15 \times 10^{-13}$ | -0.0068 | $1.19 \times 10^{-8}$ | No |
| rs2439823 | 10 | 99,778,226 | CRTAC1 | Intronic | A/G | 0.45 | 0.0072 | $1.12 \times 10^{-8}$ | 0.0036 | $2.92 \times 10^{-3}$ | No |
| rs10883618 | 10 | 103,117,653 | BTRC | Intronic | G/A | 0.37 | -0.0082 | $4.10 \times 10^{-10}$ | -0.0063 | $4.11 \times 10^{-7}$ | No |
| rs4109292 | 10 | 126,710,654 | CTBP2 | Intronic | G/A | 0.49 | -0.0073 | $7.82 \times 10^{-9}$ | -0.0049 | $4.02 \times 10^{-5}$ | No |
| rs11039324 | 11 | 47,665,686 | MTCH2 | Intergenic | G/A | 0.40 | 0.0100 | $3.94 \times 10^{-15}$ | 0.0053 | $1.44 \times 10^{-5}$ | No |
| rs10750025 | 11 | 113,424,042 | DRD2 | Intergenic | C/T | 0.32 | 0.0086 | $2.03 \times 10^{-10}$ | 0.0075 | $6.99 \times 10^{-9}$ | No |
| rs10862220 | 12 | 81,430,599 | ACSS3 | Intronic | T/G | 0.32 | -0.0084 | $2.50 \times 10^{-10}$ | -0.0072 | $1.09 \times 10^{-8}$ | No |
| rs6539771 | 12 | 84,077,443 | TMTC2 | Intronic | C/T | 0.36 | 0.0081 | $7.51 \times 10^{-10}$ | 0.0065 | $2.33 \times 10^{-7}$ | No |
| rs61954974 | 12 | 123,074,169 | KNTC1 | Intronic | T/C | 0.26 | 0.0087 | $1.76 \times 10^{-9}$ | 0.0045 | $9.76 \times 10^{-4}$ | No |
| rs12883788 | 14 | 33,303,540 | AKAP6 | Intergenic | C/T | 0.46 | 0.0074 | $2.43 \times 10^{-9}$ | 0.0041 | $5.03 \times 10^{-4}$ | No |
| rs8005131 | 14 | 33,591,105 | NPAS3 | Intronic | G/C | 0.34 | -0.0073 | $4.69 \times 10^{-8}$ | -0.0076 | $2.05 \times 10^{-9}$ | Yes |
| rs8010773 | 14 | 46,956,863 | RPL10L | ncRNA | T/C | 0.38 | 0.0078 | $1.12 \times 10^{-9}$ | 0.0051 | $2.89 \times 10^{-5}$ | No |
| rs45583845 | 14 | 57,858,194 | NAA30 | Exonic | C/G | 0.03 | 0.0205 | $1.03 \times 10^{-8}$ | 0.0182 | $1.02 \times 10^{-7}$ | Yes |
| rs8011870 | 14 | 80,173,397 | NRXN3 | ncRNA | G/A | 0.29 | 0.0077 | $3.16 \times 10^{-8}$ | 0.0063 | $2.48 \times 10^{-6}$ | Yes |
| rs7492565 | 14 | 100,985,577 | WDR25 | Intronic | G/T | 0.39 | 0.0078 | $9.37 \times 10^{-10}$ | 0.0076 | $7.53 \times 10^{-10}$ | No |
| rs1636600 | 15 | 75,609,488 | ANP32BP1 | Intergenic | G/A | 0.13 | 0.0122 | $9.73 \times 10^{-11}$ | 0.0107 | $2.74 \times 10^{-9}$ | Yes |

**Table 1 (continued)**

| SNP | Chr | Position | Implicated gene | Function | EA/NEA | MAF | Self-reported walking pace | | Self-reported walking pace adjusted for BMI | | Novel |
|---|---|---|---|---|---|---|---|---|---|---|---|
| | | | | | | | Beta | P-value | Beta | P-value | |
| rs7187776 | 16 | 28,857,645 | TUFM | UTR5 | A/G | 0.40 | 0.0094 | $1.80 \times 10^{-13}$ | 0.0044 | $3.32 \times 10^{-4}$ | No |
| rs34898535 | 16 | 31,025,641 | STX1B | Intergenic | C/T | 0.38 | −0.0071 | $3.30 \times 10^{-8}$ | −0.0023 | $6.30 \times 10^{-2}$ | No |
| rs9972653 | 16 | 53,814,363 | FTO | Intronic | G/T | 0.40 | 0.0103 | $8.58 \times 10^{-16}$ | −0.0039 | $1.19 \times 10^{-3}$ | No |
| rs4516268 | 17 | 1,846,831 | RTN4RL1 | Intronic | C/A | 0.19 | −0.0095 | $2.44 \times 10^{-9}$ | −0.0051 | $8.40 \times 10^{-4}$ | No |
| rs2301597 | 17 | 43,173,273 | NMT1 | Intronic | T/C | 0.42 | −0.0076 | $2.72 \times 10^{-9}$ | −0.0063 | $2.49 \times 10^{-7}$ | No |
| rs376942435 | 17 | 47,112,117 | IGF2BP1 | Intronic | A/C | 0.30 | −0.0084 | $6.26 \times 10^{-10}$ | −0.0046 | $4.36 \times 10^{-4}$ | No |
| rs1652376 | 18 | 21,109,466 | NPC1 | UTR3 | G/T | 0.46 | −0.0084 | $2.54 \times 10^{-11}$ | −0.0044 | $2.43 \times 10^{-4}$ | No |
| rs2469878 | 18 | 38,240,381 | PIK3C3 | Intergenic | C/T | 0.33 | 0.0074 | $3.59 \times 10^{-8}$ | 0.0062 | $1.41 \times 10^{-6}$ | Yes |
| rs784257 | 18 | 53,397,199 | TCF4 | ncRNA | T/C | 0.19 | 0.0134 | $1.03 \times 10^{-16}$ | 0.0101 | $4.81 \times 10^{-11}$ | No |
| rs67625472 | 19 | 4,968,620 | KDM4B | Intergenic | C/T | 0.28 | 0.0077 | $4.22 \times 10^{-10}$ | 0.0070 | $1.81 \times 10^{-7}$ | No |
| rs273512 | 19 | 18,224,729 | MAST3 | Intronic | T/A | 0.40 | 0.0095 | $6.91 \times 10^{-14}$ | 0.0068 | $2.33 \times 10^{-8}$ | No |
| rs1881338 | 19 | 18,838,014 | CRTC1 | Intronic | G/A | 0.49 | −0.0074 | $3.68 \times 10^{-9}$ | −0.0041 | $6.30 \times 10^{-4}$ | No |
| rs12461902 | 19 | 30,265,235 | CCNE1 | Intergenic | A/C | 0.33 | 0.0082 | $1.13 \times 10^{-9}$ | 0.0045 | $5.52 \times 10^{-4}$ | No |
| rs1667369 | 19 | 37,489,617 | ZNF568 | Intergenic | A/G | 0.37 | 0.0073 | $2.72 \times 10^{-8}$ | 0.0061 | $7.74 \times 10^{-7}$ | No |
| rs35741895 | 19 | 47,982,462 | KPTN | Intronic | A/G | 0.12 | 0.0107 | $2.30 \times 10^{-8}$ | 0.0104 | $1.48 \times 10^{-8}$ | Yes |
| rs143384 | 20 | 34,025,756 | GDF5 | UTR5 | A/G | 0.40 | −0.0076 | $1.68 \times 10^{-9}$ | −0.0075 | $7.56 \times 10^{-10}$ | No |

*P*-value of Wald test of association from BOLT-LMM mixed model analysis (*N* = 450,967)
*Chr* chromosome, *Position* hg19, *Implicated gene* nearest gene based on positional mapping, *ncRNA* non-coding RNA, *EA* effect allele, *NEA* non-effect allele, *MAF* minor allele frequency

To prevent against the potential pleiotropic effects of adiposity-related SNPs in the gene analysis, we further assessed genes that remained prioritised following adjustment for BMI. Of the 152 genes implicated by both the gene mapping and gene-based analyses, 78 remained significantly associated with self-reported walking pace following BMI adjustment. These genes included *GDF5*, *ACBD4*, *H1FX*, *PTPN9*, *FAM83C* and *UQCC1* which have previously been associated with height[17–20]; *MMP24*, *NCOA6*, *PIGU*, *GSS* and *PLCD3*, associated with lean body mass[21,22]; *MAPT*, *TRPC4AP*, *DCAKD*, *GGT7* and *PROCR*, associated with heel bone mineral density[23], and several genes linked to educational attainment[24] and cognitive function[25] (*SDCCAG8*, *BTN3A2*, *TCF4*, *HIST1H4H*, *ABT1*, *TXNL1*, *NYAP2* and *ZNF322*).

We assessed whether tissue types from the GTEx database[26] were enriched for differences in self-reported walking pace. Genes that were associated with self-reported walking pace had increased expression in the brain ($P = 9.6 \times 10^{-4}$) and pituitary ($P = 3.1 \times 10^{-6}$), with tissue-specific enrichments found in the cerebellar hemisphere ($P = 5.4 \times 10^{-7}$) and cerebellum ($P = 2.1 \times 10^{-6}$) (Supplementary Data 7).

**Interpretable SNP-heritability estimates**. To provide an interpretable heritability estimate for an ordinal outcome, we parameterised self-reported walking pace on the liability scale. Self-reported walking pace on the observed scale *y* takes the values 0, 1 and 2 with frequencies $\pi_j$ for the three ordered categories. The underlying latent variable $l \sim N(0, 1)$ is related to the observed scale through thresholds $t_1$ and $t_2$ in the equation

$$y = 1\{l > t_1\} + 1\{l > t_2\}.$$

The heritabilities on the observed and liability scales are related using the result

$$h_l^2 = h_o^2 \frac{V_0}{(z_1 + z_2)^2} \qquad (1)$$

derived by Gianola[27], where $z_j$ is the standard normal density at threshold $t_j$ and $V_o = \sum_{k=1}^{3} k^2 \pi_k - (\sum_{k=1}^{3} k\pi_k)^2$ (see Supplementary Note 1).

We used BOLT-REML[28] adjusting for age, sex, genotyping array and 20 principal components to first estimate the SNP-heritability on the observed scale. Then, using Eq. (1) to convert between scales, we estimated the SNP-heritability for self-reported walking pace on the liability scale as 13.2% (s.e. = 0.21%). With BMI included as a covariate, the heritability is reduced to 8.9% (s.e. = 0.17%).

**Genetic overlap with other traits and diseases**. We assessed whether self-reported walking pace has a shared genetic basis with other complex traits, which may reflect common biological mechanisms or causal effects in either direction. We examined genetic correlations $r_g$ between self-reported walking pace and a range of 53 traits using LD score regression[13]. The traits were assorted into categories including anthropometric traits, cardiometabolic traits, cognition and educational attainment, and aging-related traits. We observed significant genetic correlations with 39 traits based on a Bonferroni corrected threshold ($P < 9.4 \times 10^{-4}$), with results summarised in Fig. 2 and Supplementary Data 8.

The genetic architecture of self-reported walking pace overlaps highly with traits relating to adiposity (BMI, $r_g = -0.52$, $P = 4.7 \times 10^{-179}$), education and cognition (years of schooling, $r_g = 0.51$, $P = 3.4 \times 10^{-170}$; intelligence $r_g = 0.34$, $P = 3.1 \times 10^{-72}$) and longevity (parents' age at death, $r_g = 0.54$, $P = 3.9 \times 10^{-12}$). Overall, traits related to cardiometabolic risk, lung function,

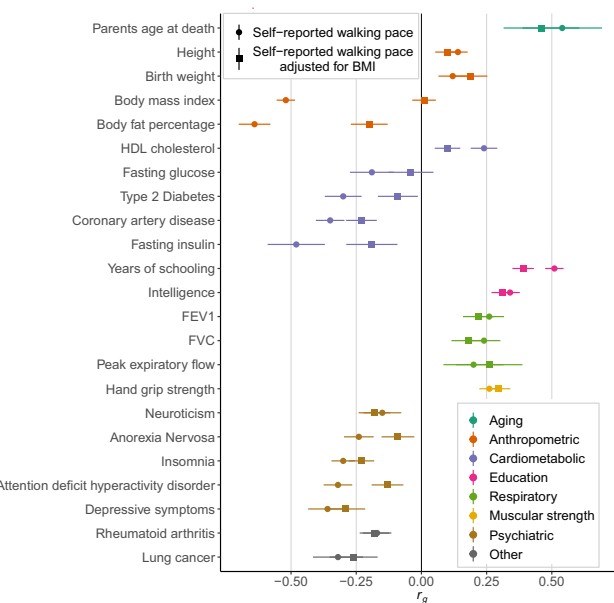

**Fig. 2 Summary of significant genetic correlations between self-reported walking pace and other phenotypes.** $r_g$, genetic correlation estimated by LD score regression. Horizontal bars represent 95% confidence intervals for the $r_g$ estimates. A Bonferroni threshold was used to test 53 phenotypes ($P < 9.4 \times 10^{-4}$). Complete results are shown in Supplementary Data 8.

psychiatric disease and muscular strength show genetic correlations with self-reported walking pace. The genetic correlations also support many of the phenotypic associations that have been observed across categories of walking pace in external cohorts[29,30]. Traits that remained genetically correlated with self-reported walking pace after adjusting for BMI included hand grip strength, measures of lung function such as forced vital capacity (FVC) and forced expiratory volume in 1 s (FEV1), years of schooling, intelligence, insomnia and depressive symptoms. Genetic correlations with adiposity-related traits and glycemic traits were attenuated following adjustment for BMI.

**Polygenic risk score association with all-cause mortality**. We explored whether the strong associations that exist between self-reported walking pace and survival[2] can be explained partly through genetic predisposition. Cox proportional hazard models were used to test the association of genetically predicted walking pace, estimated through polygenic risk scores (PRS) with a range of $P$-value thresholds, and all-cause mortality. We conducted our analyses using sex-stratified GWAS results for self-reported walking pace (see "Methods") to control for sample overlap.

We observed a significant association between genetic variants associated with self-reported walking pace and all-cause mortality in males (PRS with $P < 10^{-2}$; hazard ratio (HR) = 0.95; 95% CI: 0.92–0.97; $P = 1.93 \times 10^{-5}$) and females (PRS with $P < 10^{-2}$; HR = 0.95; 95% CI 0.92–0.98; $P = 2.70 \times 10^{-3}$) (Table 2). We performed further analyses to examine the possibility of BMI acting as a mediator of the associations. When we adjusted for BMI in the model, the association with all-cause mortality remained significant both in males (PRS with $P < 10^{-2}$; HR = 0.96; 95% CI 0.93–0.98; $P = 4.40 \times 10^{-4}$) and females (PRS with $P < 10^{-2}$; HR = 0.95; 95%CI 0.92–0.98; $P = 2.24 \times 10^{-3}$), which suggests the effect of the genetic variants on mortality is partly independent of BMI.

**Mendelian randomisation**. We performed MR to test for credible causal associations between walking pace and genetically

correlated traits. We tested 21 traits for causal relationships with self-reported walking pace at a Bonferroni significance threshold of $P < 2.3 \times 10^{-3}$, using only GWAS data from large scale, published studies of European ancestry that do not include participants from the UK Biobank cohort. The 75 lead SNPs for self-reported walking pace were used as genetic instruments within a two-sample MR, with walking pace as the exposure.

Genetically predicted self-reported walking pace was associated with a range of cardiometabolic, respiratory, psychiatric and sleeping traits (Supplementary Data 9). An increase in genetically predicted walking pace is associated with lower BMI ($\beta_{IVW} = -1.37$, $P_{IVW} = 6.7 \times 10^{-12}$), lower risk of coronary artery disease (odds ratio (OR) = 0.34, $P_{IVW} = 3.1 \times 10^{-8}$), higher HDL cholesterol levels ($\beta_{IVW} = 0.95$, $P_{IVW} = 3.3 \times 10^{-9}$) and higher FEV1 ($\beta_{IVW} = 0.35$, $P_{IVW} = 2.9 \times 10^{-5}$). We found no evidence of directional pleiotropy by testing the intercept of MR-Egger analysis (Supplementary Data 9).

To examine the potential pleiotropic effects of adiposity-related SNPs on the MR results, we conducted two sensitivity analyses accounting for the effects of BMI.

Firstly, largely similar results were found when we excluded 28 SNPs that were previously associated with BMI (Supplementary Data 10). Similar magnitude associations remained between increased genetically predicted walking pace and lower risk of coronary artery disease (OR = 0.37, $P_{IVW} = 1.5 \times 10^{-5}$) and higher FEV1 ($\beta_{IVW} = 0.33$, $P_{IVW} = 1.6 \times 10^{-3}$), though a weaker effect was observed on lowering BMI ($\beta_{IVW} = -0.55$, $P_{IVW} = 2.1 \times 10^{-6}$). Associations were substantially weakened following the exclusion of adiposity-related SNPs between genetically predicted walking pace and glycemic traits such as fasting insulin, HOMA-IR (homeostasis model assessment of insulin resistance index) and type 2 diabetes, suggesting a contribution of pleiotropy that confounds the MR results in these cases.

Secondly, we included both self-reported walking pace and BMI in a multivariable MR (Supplementary Data 11). After the inclusion of BMI as an exposure, only the association between genetically predicted walking pace and waist-to-hip ratio remained significant. This may suggest that the observed associations found between genetically predicted walking pace on lower cardiovascular risk and improved lung function are pleiotropically mediated through BMI. Alternatively, because the multivariable MR tests the direct causal effect of walking pace while holding BMI constant, the analysis may have limited power to detect such an effect when the causal effect of walking pace is substantially mediated through BMI.

## Discussion

We present a GWAS of self-reported walking pace using data from 450,967 individuals of European ancestry in the UK Biobank cohort. We identified 70 independent genomic loci associated with self-reported walking pace, of which 59 have previously reported associations in published GWAS for other traits and diseases, and 11 are currently unique to self-reported walking pace.

We estimated the SNP-heritability of self-reported walking pace as 13.2% on the liability scale, showing only a modest genetic component, suggesting that self-reported walking pace is largely modifiable. We showed that there are many significant genetic correlations with cardiometabolic traits and diseases, including BMI, coronary heart disease, type 2 diabetes and lipid levels, with respiratory traits and other lifestyle behaviours such as sleep. These could be due either to causal associations between self-reported walking pace and those traits, in either direction, or through pleiotropic effects whereby genetic variants influence

**Table 2 Association between genetically determined self-reported walking pace and all-cause mortality, stratified by sex. PRS, polygenic risk score. Hazard ratios are per 1 standard deviation increased PRS.**

| Association with mortality | Number of SNPs | Hazard ratio | 95% CI | P-value |
|---|---|---|---|---|
| *Males (N = 186,015)* | | | | |
| PRS ($P < 10^{-2}$) | 24,982 | 0.95 | 0.92–0.97 | $1.93 \times 10^{-5}$ |
| PRS ($P < 10^{-3}$) | 4280 | 0.96 | 0.93–0.98 | $3.36 \times 10^{-4}$ |
| PRS ($P < 10^{-4}$) | 841 | 0.96 | 0.94–0.98 | $1.64 \times 10^{-3}$ |
| PRS ($P < 5 \times 10^{-5}$) | 523 | 0.96 | 0.93–0.98 | $4.65 \times 10^{-4}$ |
| PRS ($P < 8 \times 10^{-8}$) | 16 | 0.98 | 0.96–1.01 | 0.17 |
| PRS ($P < 10^{-2}$) with adj. for BMI | 24,982 | 0.96 | 0.93–0.98 | $4.40 \times 10^{-4}$ |
| *Females (N = 223,646)* | | | | |
| PRS ($P < 10^{-2}$) | 23,851 | 0.95 | 0.92–0.98 | $2.70 \times 10^{-3}$ |
| PRS ($P < 10^{-3}$) | 3831 | 0.96 | 0.93–0.99 | $1.51 \times 10^{-2}$ |
| PRS ($P < 10^{-4}$) | 689 | 0.95 | 0.92–0.98 | $1.16 \times 10^{-3}$ |
| PRS ($P < 5 \times 10^{-5}$) | 420 | 0.95 | 0.92–0.98 | $1.26 \times 10^{-3}$ |
| PRS ($P < 8 \times 10^{-8}$) | 12 | 0.97 | 0.94–1.00 | $3.27 \times 10^{-2}$ |
| PRS ($P < 10^{-2}$) with adj. for BMI | 23,851 | 0.95 | 0.92–0.98 | $2.24 \times 10^{-3}$ |

multiple phenotypes through possibly independent biological pathways[31]. We showed also that polygenic scores predicting self-reported walking pace are inversely associated with all-cause mortality risk, and this association is independent of BMI. Future work examining the genetic relationship between walking pace and survival could focus on the biological mechanisms underlying these associations.

By performing MR analyses we provide evidence that a genetically elevated self-reported walking pace is linked to a lower cardiometabolic risk profile, suggesting that increasing walking pace could act as a beneficial intervention for a range of health outcomes. This is consistent with findings from randomised controlled trials in cardiovascular disease patients, which have shown that exercise-based interventions have beneficial effects on survival[32]. Our results suggest that such interventions may also be effective in the general population of adults. MR depends upon a number of assumptions to draw causal inferences, with many methods available to vary the required assumptions[31]. An exhaustive analysis of every causal association is beyond the scope of this study, but we have allowed for the impact of pleiotropy with MR-Egger and weighted median methods, and further sensitivity analyses to examine the effect of adiposity-related SNPs. As self-reported walking pace is a general indicator of an individual's perceived health, there are likely to be many different biological and psychological mechanisms underlying it. The specific mechanisms are unclear though, as is the extent to which they might invalidate the MR results. By using a range of MR estimators, which depend on different, though related sets of assumptions, we can increase the reliability of our causal inferences. We believe that the ensemble of significant MR results across phenotypes, with effects in biologically plausible directions, is sufficient to conclude with confidence that increasing self-reported walking pace would cause certain aspects of health to improve, and thus is likely to be a suitable target for intervention. In addition, because the phenotype is a self-reported measure, our results may also support a causal link between positive self-perceptions of health and overall health status.

To better understand the relationship between self-reported walking pace and BMI we performed several sensitivity analyses. The high genetic correlation between self-reported walking pace with and without adjustment for BMI ($r_g = 0.83$, s.e. = 0.0073) suggests a substantial component of the genetic architecture of self-reported walking pace is independent of BMI. This was supported by genetic correlations between self-reported walking pace and a range of complex traits and diseases that were largely robust to adjustment for BMI. In comparison with the genome-wide correlations, a more marked effect of BMI was noted at the genomic loci associated with self-reported walking pace. Only 15 of the 70 loci survived the adjustment for BMI at the genome-wide significance level, whilst 45 loci in total retained a suggestive level of $P < 10^{-5}$. The attenuation of top hits may partly reflect a mediated effect of BMI on the causal pathway between genotype and self-reported walking pace. To explore this, we performed multivariable MR which is a valid form of mediation analysis[33]. Following the inclusion of BMI as a secondary exposure alongside self-reported walking pace, we found that across a range of outcomes there was weak evidence of an indirect causal effect (independent of BMI) of self-reported walking pace. One possible explanation to note however for this finding is the limited statistical power available to accurately detect both direct and indirect causal effects in a multivariable MR setting.

We found that a self-reported walking pace has a strong genetic overlap with increased years in education and greater intelligence. Hypotheses have been proposed to explain the association between walking pace and both educational and cognitive outcomes[34]. Firstly, educational attainment may be associated with positive lifestyle choices regarding physical activity and diet, and in addition, a higher education is associated with a greater ability to self-manage health such as by using health services effectively. The importance of walking pace as a measure of overall health status is further supported by previous evidence showing this phenotype is correlated highly with objective measures of physical fitness[1]. A faster walking pace may also reflect psychological factors relating to increased motivation and internal "drive", which are plausibly linked to educational attainment and cognition. In addition, it has been observed that in old age there is a parallel decline of walking pace and cognition, and our results may provide some evidence of a genetic basis to this association. Future work could explore this further through joint analysis of walking pace and age-related neurological diseases associated with loss of cognition.

A strong genetic correlation was also observed between self-reported walking pace and hand grip strength, a proxy for overall muscle strength[35]. In addition, 13 genome-wide significant loci for hand grip strength overlapped with our 70 self-reported walking pace loci. Similar to walking pace, hand grip strength is known to decline with age, whilst increasing muscular strength has been shown to improve functional capacity[36]. These results indicate a shared genetic basis to the associations that both hand grip strength and walking pace display towards age-related phenotypes. There is however potential for pleiotropic effects that act through the same genetic variants on distinct biological pathways,

and further work is needed on the biological mechanisms relating to the self-reported walking pace loci to understand their relevance to muscular strength.

Further work may also include bidirectional MR analyses and mediation analyses to understand the relative importance of walking pace and adiposity on health and survival outcomes. The release of detailed data acquired by accelerometer devices on a subset of participants[37] presents further opportunities to compare self-reported walking pace with objective measures of physical activity at both a phenotypic and genotypic level.

Our analysis revealed challenges that are introduced when analysing an ordered categorical phenotype. Rather than the classical modelling approach of an ordinal logistic regression, we assigned weights to the ordered categories and used a linear mixed model. The linear scale makes strong assumptions about the distances between the categories of self-reported walking pace. Whilst recently developed ordinal logistic regression methods have been applied to non-imputed data at UK Biobank scale[38], they are not yet computationally tractable on densely imputed GWAS datasets. Analysing ordered categorical variables on the linear scale proves problematic when interpreting SNP effect sizes, SNP-heritability and causal effect estimates in MR. We converted heritability estimates from the observed scale to the liability scale, which is more interpretable as it models self-reported walking pace as a continuous trait. This unobserved latent scale is not the actual walking pace, which can be measured under controlled conditions[7], but reflects genetic and environmental factors that influence the self-reported category.

There are several limitations to note. First, the associated loci must be accepted tentatively until validated in an independent cohort. We were specifically interested in the self-reported phenotype owing to its ease of measurement, but while similar measures are available in some prospective cohorts, we were unable to obtain the relevant data during the course of this study. In particular, it is important to confirm the results in a separate demographic, since the UK Biobank participants are known to be healthier than the general population[39]. Second, self-reported walking pace is known to be a crude measure in comparison to objective assessments, which raises the possibility of misclassification bias[40]. In particular, it is thought that self-reported walking pace reflects both actual walking pace in daily life as well as a sense of self-rated health[41,42]. Nonetheless, previous studies have indicated a reasonably close association exists between self-reported and objectively measured usual walking pace[43,44], and work by Murtagh et al.[45] showed that issuing a simple instruction to walk "briskly" prompted more vigorous activity in participants across all fitness levels. Third, this work is limited in scope by the lack of questionnaire data on the specific context of the walking behaviour, as walking pace is known to differ across domains (e.g. exercise, travel, domestic, leisure)[46].

Therefore, the genetic associations and possible causal effects we report here may not hold for more specific measures of gait. Nevertheless, the strong association of self-reported walking pace with health outcomes and mortality warrants study in its own right. Despite the inherent limitations described, our results highlight the value of studying subjective, self-reported measures of physical activity. We are able to utilise a simple measure of self-reported walking pace to infer that walking at a speed that is brisk in one's own estimation has important benefits to health and longevity. Arguably this could provide the basis for health advice that is easier to understand and follow compared to walking at or above a precisely defined speed. Nevertheless, further investigation is needed into the generalisability of our findings to interventions aimed at increasing objectively assessed walking pace.

In conclusion, we have identified 70 genetic loci associated with self-reported walking pace and shown that its strong associations with cardiorespiratory and mortality outcomes is partly explained by genetic correlations. MR arguments augment the results of trials on cardiovascular patients[32] to suggest that self-reported walking pace may be a beneficial target for intervention in the general population. Given its ease of measurement, by definition by individuals themselves, it may be entirely feasible to develop pragmatic interventions on walking pace that have beneficial effects on health.

## Methods

**Study population**. The UK Biobank study is a large cohort of 501,726 British residents aged between 40 and 69 at recruitment. The participants attended assessment visits across 23 study centres in the UK, through which extensive phenotypic data were collected. Participants provided informed consent to participate, and the UK Biobank study has ethics approval from the North West National Research Ethics Committee (REC reference 11/NW/0274). This work has been conducted under UK Biobank application 33266.

**Genotype, imputation and quality control**. The initial genotyping, imputation and quality control were conducted centrally by the UK Biobank, and have been described in detail elsewhere[9]. Genotyping was performed using the UK BiLEVE Axiom Array and the UK Biobank Axiom arrays, with imputation to the Haplotype Reference Consortium panel[47] which has approximately 96 million variants.

**Phenotype**. Self-reported walking pace was ascertained using the ACE touchscreen question "How would you describe your usual walking pace?" with response options of "slow", "steady/average", "brisk", "None of the above" or "Prefer not to answer". If the participant activated the "Help" button they were shown the message: "Slow pace is defined as less than 3 miles per hour. Steady average pace is defined as between 3-4 miles per hour. Fast pace is defined as more than 4 miles per hour." We excluded participants whose answers were "None of the above" ($n = 1,426$) or "Prefer not to answer" ($n = 519$). The low numbers of these exclusions suggest minimal impact of any informative missingness. The responses "slow", "steady/average" and "brisk" were coded as 0, 1 and 2 for our analyses.

**Genome-wide association analysis**. Association analysis was carried out in a set of 450,967 individuals of European ancestry with non-missing phenotypes, where ancestry was defined by the K-means clustering of the first two principal components[48]. A linear mixed non-infinitesimal model for self-reported walking pace was implemented in BOLT-LMM v2.3.2[10] under an additive genetic model. The model included covariates for age, sex, genotyping array and the first 20 principal components of ancestry. We additionally carried out a sensitivity analyses to explore the effect of using BMI as a covariate. Following association analysis, only biallelic SNPs were retained with a minor allele frequency ≥0.005, imputation quality ≥0.60 and maximum per SNP missingness of 10%. In total, 10,061,374 variants were analysed. To estimate the linear mixed model parameters further QC was performed to remove variants with a minor allele frequency <1% and deviation from Hardy-Weinberg equilibrium ($P < 10^{-6}$).

Genomic risk loci were derived using the Functional Mapping and Annotation of genetic associations (FUMA) platform[49]. Independent significant SNPs were defined using a genome-wide significance threshold of $P < 5 \times 10^{-8}$, independent from each other at $r^2 < 0.6$. Lead SNPs were further identified as a subset of the independent significant SNPs that are in linkage disequilibrium (LD) at $r^2 < 0.1$. Genomic loci were defined by merging lead SNPs that are located within 250 kb of each other.

Interaction effects for the lead SNPs by sex were investigated by carrying out the BOLT-LMM analyses stratified by sex. The strata were ensured to be approximately independent by excluding individuals related to 3rd degree or above (kinship coefficient <0.044) using the software KING[50]. In each 3rd degree related pair, we retained the individual with the lower genotyping missingness rate.

The effect of confounding by population structure was estimated using the intercept of the LD score regression, which estimates the inflation in test statistics due to confounding of the association between walking pace and genotype[13].

**Sensitivity analysis**. Because we used a linear model to test association with an ordinal categorical trait, we assessed the sensitivity of the results to different coding schemes of the self-reported walking pace phenotype, and compared statistical power when using an ordinal logistic and linear model. We partitioned the GWAS SNPs into 6 minor allele frequency bins where we randomly selected 1000 SNPs from each, and compared the $P$-value of association for these SNPs under both the linear and ordinal logistic models (Supplementary Fig. 1). We additionally compared SNP effect sizes under both the linear and ordinal logistic models for the 75 independent lead SNPs (Supplementary Fig. 2). We used a sample of 373,414 unrelated individuals, such that no pair are related to 3rd degree or above, corresponding to a KING kinship coefficient[50] of <0.044. We fitted both linear and ordinal logistic models with covariates for age, sex, genotyping array and

20 principal components using PLINK1.9[51] for the linear model and the Julia package *OrdinalGWAS.jl*[38] for the ordinal logistic model.

**Genetic correlations**. The genetic correlations $r_g$ between self-reported walking pace and 53 traits were estimated using LD Score regression performed through the LDSCv1.0.1 software[13]. The set of traits includes anthropometric, cardiometabolic, educational, bone mineral density, aging and other categories for which summary statistic data was publicly available. Genetic correlations were tested for significance using a Bonferroni correction of $P < 9.4 \times 10^{-4}$.

**Post-GWAS annotation and functional mapping**. The functional annotation of SNPs associated with self-reported walking pace was carried out using FUMA[49]. Annotations include ANNOVAR categories, CADD scores, RegulomeDB scores and chromatin states. All candidate SNPs in the genomic risk loci (SNPs with $r^2 \geq 0.6$ with the lead SNPs and a suggestive significance level $P < 5 \times 10^{-5}$) were annotated.

Positional mapping and eQTL mapping were used to link self-reported walking pace genomic loci to genes. We used the prioritised genes from the positional and eQTL mapping to perform gene-set enrichment analysis against gene sets defined by traits in the GWAS catalogue. Additionally, gene-based analysis was performed with MAGMA through the FUMA platform[52]. MAGMA combines the P-values for SNPs within a gene to create gene-based P-values for 19,834 protein-coding genes. A Bonferroni corrected threshold of $P < 2.52 \times 10^{-6}$ was used to determine significantly associated genes. Finally, we used FUMA to perform tissue-enrichment analysis of 30 broad tissue types and 54 specific tissue types from the GTEx database[26].

**GWAS catalogue lookup**. We identified SNPs with previously reported ($P < 10^{-5}$) phenotypic associations in published GWAS in the NHGRI-EBI catalogue which overlap with SNPs in LD ($r^2 > 0.6$) with the independent significant SNPs.

**Polygenic risk score association with all-cause mortality**. Cox proportional hazard models were used to investigate the association between genetically determined self-reported walking pace with all-cause mortality, using age as time scale. Analyses were stratified by sex. For males there were 7049 all-cause mortality cases ($n$ total $= 186,015$) and for females 4546 cases ($n$ total $= 223,646$). To test for association with all-cause mortality in males, we computed genetic risk scores weighted by effect sizes estimated from the independent sample of females, and vice versa. The polygenic risk scores were constructed using PRSice v2.2.3 software[53] for a range of P-value thresholds between $5 \times 10^{-8}$ and $10^{-2}$, using approximately independent genetic markers obtained by clumping the SNPs with an $r^2$ threshold of 0.1 and a window size of 250 kb. To examine the robustness of these associations to adiposity as a mediator, we included covariate adjustment for BMI.

Analyses were performed with Stata 16.0. Mortality status was obtained from the UK Biobank through the National Health Service (NHS) Information Centre and the NHS Central Register, Scotland with detailed information on the data linkage procedure available online.

**MR analyses**. To investigate whether walking pace has a causal effect on different outcomes, we performed two-sample MR analyses testing 21 traits identified in the genetic correlation analysis. We used only GWAS data from large scale, previously published studies of European ancestry that do not include participants from the UK Biobank cohort. The inverse variance weighted approach was used as the primary method to infer causal effect estimates. The potential effect of pleiotropy was evaluated using the MR-Egger and weighted median estimate methods[54,55]. MR-Egger requires the InSIDE assumption to hold (Instrument Strength Independent of Direct Effect), whilst the weighted median approach requires no more than 50% of the weighted instruments to be invalid due to horizontal pleiotropy. The 75 independent lead SNPs were used as instrumental variables, using proxies in strong LD ($r^2 > 0.80$) if the SNPs were unavailable in the outcome GWAS.

We conducted further sensitivity analyses to explore the effect of pleiotropy due to BMI, as several of the SNP associations for self-reported walking pace are shared with BMI. Firstly, we conducted the MR analyses with the 28 lead SNPs previously associated with BMI excluded. Secondly, we performed multivariable MR by including both self-reported walking pace and BMI as exposures[56]. Estimates in this case correspond to the direct causal effect of walking pace with BMI being fixed. The summary statistic data on BMI was obtained from The Genetic Investigation of Anthropometric Traits (GIANT) consortium[12].

MR analyses were performed using the MendelianRandomisation[57] package implemented in R software.

**Reporting summary**. Further information on research design is available in the Nature Research Reporting Summary linked to this article.

## Data availability

The GWAS summary statistics for self-reported walking pace are available via Figshare at https://doi.org/10.6084/m9.figshare.12967088.v1[58]. The GWAS summary statistics for self-reported walking pace, adjusted for BMI, are available via Figshare at https://doi.org/10.6084/m9.figshare.12967091.v1[59]. Individual-level genotype data are available by application to the UK Biobank.

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

## Acknowledgements

This research has been conducted using the UK Biobank Resource under application number 33266. TY is supported by the Medical Research Council (MR/T031816/1) and the NIHR Leicester Biomedical Research Centre.

## Author contributions

I.R.T. developed methods, performed analyses, interpreted results and wrote the manuscript. F.Z. conceived the study, interpreted results and edited the manuscript. C.P.N. interpreted results and edited the manuscript. P.W.F. interpreted results and edited the manuscript. T.Y. conceived the study, interpreted results and edited the manuscript. F.D. conceived the study, interpreted results and edited the manuscript.

## Competing interests

The authors declare no competing interests.
