## [Peer Review File · Communications Biology]

Reviewers' comments:

Reviewer #1 (Remarks to the Author):

To the authors: I have been asked by the editor to review the parts of the manuscript relating to walking pace as this is my area of expertise as a movement scientist focusing on gait. Therefore, my review focuses only on these parts of the manuscript.

My primary concerns relate to the fact that walking pace was self-reported and the limitations of this are not stated or discussed as clearly as they perhaps should be. In my field, it is quite well accepted that self-reported measures of walking speed and physical activity in general are very crude measures. Even some simple objective measures of walking speed have recently shown to lack validity as measures of gait in daily life (e.g. <https://doi.org/10.1016/j.maturitas.2018.12.008>).

In the question posed to the participants in the current study, it is unclear how the option they select relates to their typical or most common daily life walking behaviour and the extent to which they may under or overestimate their own activity and/or capacity. It is well established that people walk at a range of different speeds throughout the day depending on the type and duration of tasks being undertaken. If there is any data on these issues in relation to the questionnaire used, it would be helpful if this could be included in the manuscript. I realise that with the size of the dataset, much of the inefficiencies of self-reported measures will become less relevant, but nevertheless, I think it affects the study.

A related issue is whether or not perceived walking pace is likely to change in precisely the same manner as the true change in walking pace with intervention. Some of the authors' conclusions relate to the potential causal relationship between walking pace and various health outcomes, but their conclusions are based on the self-reported walking speed. Do we know that the relationship between self-reported walking speed and true walking speed is linear? It may be that a small improvement in true walking speed results in a proportionally greater improvement in perceived walking speed, or the reverse may be true. This would have implications for using self-reported vs. objectively measured walking pace in interventions.

To prevent misinterpretation, there are some key points in the manuscript where it should be clearly indicated that walking pace was assessed by self-report. I recommend adding this in a few places in the manuscript as listed below.

Title: "..self-reported walking pace.."

Line 51: "self-reported" not "habitual"

Line 57, 61 (twice), Figure 1 legends, 216, 218, 244: insert "self-reported"

As my comments above may indicate, I do not necessarily agree with the statement in lines 293-295, nor do I think a good relationship with objectively measured fitness necessarily validates self-reported walking pace as a measure of true walking pace. Ideally, more support from literature demonstrating validity should be added regarding the use of self-reported walking pace and/or the associated limitations should be discussed in more detail.

Lines 320-322 – Is it known how many participants used the help button? This may provide some insight into the intuitiveness of the question.

Reviewer #2 (Remarks to the Author):

Timmins et al., report a genome-wide association study on self-reported habitual walking pace (graded as "slow", "steady/average", "brisk" using the codes 0, 1, and 2), followed by Mendelian Randomization analyses with other biometric and health outcome information. The data allowed the authors to conclude that

- a) there are strong genetic loci associated with self-reported habitual walking pace
- b) These loci strongly associate with an improved cardiometabolic profile (i.e. higher HDL levels and lower risk of coronary arterial disease).

This is a highly important topic of research with very broad public interest, and it is performed in a large number of participants. It was a great pleasure reading this manuscript.

I have several suggestions for the authors to consider:

- a) In the main text, 70 loci surpassing genome-wide significance were found, and 59 were associated with other traits, and "21 are currently unique to self reported walking pace". Can the authors clarify whether it is 21 unique loci or 11? This can be found in the abstract and in the Discussion section, lines 215 - 216.
- b) On page 10 of the main text, lines 189 - 191: ".....lower risk of coronary artery disease (beta_{ivw} = -1.06, P_{ivw} = 5.6x10⁻⁸ 190), higher HDL cholesterol levels (beta_{ivw} = 0.95, P_{ivw} = 3.4x10⁻⁹). When I looked at Supplementary Table 9, excel file, the data reads beta_{ivw} = -1.07 and P_{ivw} = 3.1x10⁻⁸ for coronary arterial disease. For HDL cholesterol, beta_{ivw}=0.95, P_{ivw} = 3.3 x 10⁻⁹. Can the authors please examine these discrepancies? These kind of discrepancies suggest errors in reconciliation between main text and Data tables.
- c) I note, most humbly, in the reporting summary, that the authors made good-faith efforts to obtain replication data from two cohorts, but the data were not forthcoming. Looking at Supplementary Table 2, adjustment for BMI causes many loci to momentarily lose genome-wide significance. Whilst this is not fatal, the authors should discuss this more vigorously in the discussion section, as it would be of great interest to the reader. The correlation between BMI, walking pace, and health outcomes is extremely interesting regardless of how many loci reach genome-wide significance, and it is encouraging that many of the mendelian randomization analyses results are very, very significant (Supp Table 9).
- d) Putting a pubmed ID in Supplementary Table 9 may not be helpful enough to the reader. It could possibly be best to clarify further that these MR results were obtained from previously published, large scale studies, perhaps consider including samples size, etc to help your reader understand the context of the MR analysis.

The same likely applies in the main text, whereby the sentence "We performed Mendelian randomization (MR) to test for credible causal associations between walking pace and genetically correlated traits. We tested 21 traits for causal relationships with self-reported walking pace at a Bonferroni significance threshold of $\phi < 2.3 \times 10^{-3}$ 183 , using only GWAS data from studies of European ancestry that do not include participants from the UK Biobank cohort" does not provide the level of detail that can inform the reader. Perhaps consider adding in some references, and sample sizes, etc in the Supplemental data / text?

- e) For supplementary Table 2, do please consider including gene annotations to accompany the rs-ids for the SNPs.
- f) The linear mixed model analysis is wonderful. Some readers may ask whether the distance between "slow" and "steady/average" may be different between "steady/average" and "brisk". The authors justified their analysis perfectly.
- g) For case-control traits such as presence or absence of coronary arterial disease, could the authors consider providing an odds ratio (more intuitive to the reader) per-unit increase of risk alleles?

Reviewer #3 (Remarks to the Author):

This is an excellent genome-wide association study and the paper is well-written. The design was complete and the analyses were well conducted.

I have no any comments.

Referee expertise:

Referee #1: biomechanics of walking

Referee #2: GWAS, statistical genetics

Referee #3: GWAS, MR, statistical genetics

Reviewers' comments:

Reviewer #1 (Remarks to the Author):

To the authors: I have been asked by the editor to review the parts of the manuscript relating to walking pace as this is my area of expertise as a movement scientist focusing on gait. Therefore, my review focuses only on these parts of the manuscript.

My primary concerns relate to the fact that walking pace was self-reported and the limitations of this are not stated or discussed as clearly as they perhaps should be. In my field, it is quite well accepted that self-reported measures of walking speed and physical activity in general are very crude measures. Even some simple objective measures of walking speed have recently shown to lack validity as measures of gait in daily life (e.g. <https://doi.org/10.1016/j.maturitas.2018.12.008>).

Response: We agree that the limitations of a crude, self-reported measure need to be discussed in greater detail. We have added the following to the discussion:

“self-reported walking pace is known to be a crude measure in comparison to objective assessments, which raises the possibility of misclassification bias (40). In particular, it is thought that self-reported walking pace reflects both actual walking pace in daily life as well as a sense of self-rated health (41, 42) ... Therefore, the genetic associations and possible causal effects we report here may not hold for more specific measures of gait.”

It may reassure the Reviewer that in ongoing unpublished work, we are finding that the self-reported walking pace question in UK biobank is associated with the intensity of movement undertaken in the most active 30 minutes of the 24 hour cycle as assessed objectively by accelerometer (based on approx. 100,000 participants with accelerometer data in UK Biobank). Therefore it appears to be a good measure of people's intensity of habitual movement.

We do want to stress that self-reported walking pace is also of intrinsic interest both because it is strongly associated with health outcomes and mortality (possibly more so than objective measures of gait, although this is not firmly established), and because it is a pragmatic target for intervention:

“Nevertheless the strong association of self-reported walking pace with health outcomes and mortality warrants study in its own right ... this could provide the basis for health advice that is easier to understand and follow compared to walking at or above a precisely defined speed.”

In the question posed to the participants in the current study, it is unclear how the option they select relates to their typical or most common daily life walking behaviour and the extent to which they may under or overestimate their own activity and/or capacity. It is well established that people walk at a range of different speeds throughout the day depending on the type and duration of tasks being undertaken. If there is any data on these issues in relation to the questionnaire used, it would be helpful if this could be included in the manuscript. I realise that with the size of the dataset, much of the inefficiencies of self-reported measures will become less relevant, but nevertheless, I think it affects the study.

Response: As we state in the paper, the question asks “How would you describe your usual walking pace?” participants were then able to select a help button that defined “low pace is defined as less than 3 miles per hour. Steady average pace is defined as between 3-4 miles per hour. Fast pace is defined as more than 4 miles per hour.”

Therefore we believe this is interpreted as purposeful walking activity, rather than incidental tasks of daily living.

Regardless, whilst the issue of how self-reported walking pace translates to daily life walking behaviour is important and worthy of discussion, we don't believe it is a major limitation here. We argue that irrespective of how intervening on self-reported walking pace subsequently changes a given individual's daily life walking behaviours, it is predicted to have a beneficial effect on a range of different outcomes. We have added an additional sentence to the discussion:

“this work is limited in scope by the lack of questionnaire data on the specific context of the walking behaviour, as walking pace is known to differ across domains (e.g. exercise, travel, domestic, leisure) (46).”

A related issue is whether or not perceived walking pace is likely to change in precisely the same manner as the true change in walking pace with intervention. Some of the authors' conclusions relate to the potential causal relationship between walking pace and various health outcomes, but their conclusions are based on the self-reported walking speed. Do we know that the relationship between self-reported walking speed and true walking speed is linear? It may be that a small improvement in true walking speed results in a proportionally greater improvement in perceived walking speed, or the reverse may be true. This would have implications for using self-reported vs. objectively measured walking pace in interventions.

Response: We agree. As our analysis is based specifically on self-reported walking pace, our conclusions may not hold for true walking pace. Given the importance of self-reported walking pace as a risk factor for all-cause and cardiovascular mortality, we believe it is both appropriate and informative to investigate the genetic basis for this. However, we agree that caution is needed around drawing strong causal inferences around intervention designed to improve objective walking pace.

“Therefore, the ... possible causal effects we report here may not hold for more specific measures of gait ... Despite the inherent limitations described, our results highlight the value of studying subjective, self-reported measures of physical activity. We are able to utilise a simple measure of self-reported walking pace to infer that walking at a speed that is brisk in one's own estimation has important benefits to health and longevity. Arguably this could provide the basis for health advice that is easier to understand and follow compared to walking at or above a precisely defined speed. Nevertheless, further investigation is needed

into the generalisability of our findings to interventions aimed at increasing objectively assessed walking pace. “

To prevent misinterpretation, there are some key points in the manuscript where it should be clearly indicated that walking pace was assessed by self-report. I recommend adding this in a few places in the manuscript as listed below.

Title: “..self-reported walking pace..”

Line 51: “self-reported” not “habitual”

Line 57, 61 (twice), Figure 1 legends, 216, 218, 244: insert “self-reported”

Response: We agree that the self-reported nature of the assessment needs to be made clear in the title and throughout the text. We have amended the title to “GWAS of self-reported walking pace suggests beneficial effects of brisk walking on health and survival”.

Line 51, corrected to “self-reported”.

Line 57, 61 (twice): inserted “self-reported”.

Figure 1 and 2 legends: inserted “self-reported”.

Line 216, 218, 244: inserted “self-reported”.

As my comments above may indicate, I do not necessarily agree with the statement in lines 293-295, nor do I think a good relationship with objectively measured fitness necessarily validates self-reported walking pace as a measure of true walking pace. Ideally, more support from literature demonstrating validity should be added regarding the use of self-reported walking pace and/or the associated limitations should be discussed in more detail.

Response: We agree that adding some supportive literature on this point improves the discussion. We have removed the comment mentioned in lines 293-295, and have added more relevant citations from the literature to the discussion:

“previous studies have indicated a reasonably close association exists between self-reported and objectively measured usual walking pace (42, 43), and work by Murtagh et al. (44) showed that issuing a simple instruction to walk “briskly” prompted more vigorous activity in participants across all fitness levels. ”

Lines 320-322 – Is it known how many participants used the help button? This may provide some insight into the intuitiveness of the question.

Response: Unfortunately it is not known whether participants used the help button.

Reviewer #2 (Remarks to the Author):

Timmins et al., report a genome-wide association study on self-reported habitual walking pace (graded as “slow”, “steady/average”, “brisk” using the codes 0, 1, and 2), followed by Mendelian Randomization analyses with other biometric and health outcome information. The data allowed the authors to conclude that

a) there are strong genetic loci associated with self-reported habitual walking pace

b) These loci strongly associate with an improved cardiometabolic profile (i.e. higher HDL levels and lower risk of coronary arterial disease).

This is a highly important topic of research with very broad public interest, and it is performed in a large number of participants. It was a great pleasure reading this manuscript.

Response: We thank the reviewer for their overall positive assessment of the work.

I have several suggestions for the authors to consider:

a) In the main text, 70 loci surpassing genome-wide significance were found, and 59 were associated with other traits, and "21 are currently unique to self reported walking pace". Can the authors clarify whether it is 21 unique loci or 11? This can be found in the abstract and in the Discussion section, lines 215 - 216.

Response: We have changed the text to state correctly that there are 11 unique loci.

b) On page 10 of the main text, lines 189 - 191: ".....lower risk of coronary artery disease (beta_{iw} = -1.06, P_{iw} = 5.6x10⁻⁸ 190), higher HDL cholesterol levels (beta_{iw} = 0.95, P_{iw} = 3.4x10⁻⁹). When i looked at Supplementary Table 9, excel file, the data reads beta_{iw} = -1.07 and P_{iw} = 3.1x10⁻⁸ for coronary arterial disease. For HDL cholesterol, beta_{iw}=0.95, P_{iw} = 3.3 x 10⁻⁹. Can the authors please examine these discrepancies? These kind of discrepancies suggest errors in reconciliation between main text and Data tables.

Response: We have updated the manuscript with the correct effect sizes and P-values for the genetic correlations and MR results, which now coincide with the supplementary tables.

c) I note, most humbly, in the reporting summary, that the authors made good-faith efforts to obtain replication data from two cohorts, but the data were not forthcoming. Looking at Supplementary Table 2, adjustment for BMI causes many loci to momentarily lose genome-wide significance. Whilst this is not fatal, the authors should discuss this more vigorously in the discussion section, as it would be of great interest to the reader. The correlation between BMI, walking pace, and health outcomes is extremely interesting regardless of how many loci reach genome-wide significance, and it is encouraging that many of the mendelian randomization analyses results are very, very significant (Supp Table 9).

Response: We agree that the role of BMI should be discussed in more detail. Firstly, we add a further sentence to the results section:

"Of the 70 associated loci only 15 retained genome-wide significance following adjustment for BMI, whilst 45 loci in total maintained a suggestive significance level ($P < 10^{-5}$)."

And to the discussion section we add a further paragraph:

"To better understand the relationship between self-reported walking pace and BMI we performed several sensitivity analyses. The high genetic correlation between self-reported walking pace with and without adjustment for BMI ($r_g = 0.83$, $s.e. = 0.0073$) suggests a substantial component of the genetic architecture of self-reported walking pace is independent of BMI. This was supported by genetic correlations between self-reported walking pace and a range of complex traits and diseases that were largely robust to

adjustment for BMI. In comparison with the genome-wide correlations, a more marked effect of BMI was noted at the genomic loci associated with self-reported walking pace. Only 15 of the 70 loci survived the adjustment for BMI at the genome-wide significance level, whilst 45 loci in total retained a suggestive level of $P < 10^{-5}$. The attenuation of top hits may partly reflect a mediated effect of BMI on the causal pathway between genotype and self-reported walking pace. To explore this, we performed multivariable Mendelian Randomisation which is a valid form of mediation analysis (33). Following the inclusion of BMI as a secondary exposure alongside self-reported walking pace, we found that across a range of outcomes there was weak evidence of an indirect causal effect (independent of BMI) of self-reported walking pace. One possible explanation to note however for this finding is the limited statistical power available to accurately detect both direct and indirect causal effects in a multivariable MR setting."

d) Putting a pubmed ID in Supplementary Table 9 may not be helpful enough to the reader. It could possibly be best to clarify further that these MR results were obtained from previously published, large scale studies, perhaps consider including samples size, etc to help your reader understand the context of the MR analysis.

The same likely applies in the main text, whereby the sentence "We performed Mendelian randomization (MR) to test for credible causal associations between walking pace and genetically correlated traits. We tested 21 traits for causal relationships with self-reported walking pace at a Bonferroni significance threshold of $\alpha < 2.3 \times 10^{-3} / 183$, using only GWAS data from studies of European ancestry that do not include participants from the UK Biobank cohort" does not provide the level of detail that can inform the reader. Perhaps consider adding in some references, and sample sizes, etc in the Supplemental data / text?

Response: We have updated Supplementary Tables 8-11 to include columns with study reference, cohort and sample size. We have amended the methods and results section:

*"... using only GWAS data from **large scale, published** studies of European ancestry that do not include participants from the UK Biobank cohort."*

e) For supplementary Table 2, do please consider including gene annotations to accompany the rs-ids for the SNPs.

Response: We have updated Supplementary Table 2 to include a column with the nearest gene.

f) The linear mixed model analysis is wonderful. Some readers may ask whether the distance between "slow" and "steady/average" may be different between "steady/average" and "brisk". The authors justified their analysis perfectly.

g) For case-control traits such as presence or absence of coronary arterial disease, could the authors consider providing an odds ratio (more intuitive to the reader) per-unit increase of risk alleles?

Response: We have replaced betas (log odds ratios) with odds ratios for our Mendelian Randomisation analyses in both the results section and in Supplementary Tables 9-11.

Reviewer #3 (Remarks to the Author):

This is an excellent genome-wide association study and the paper is well-written. The design was complete and the analyses were well conducted.

I have no any comments.

Response: We thank the reviewer for their positive assessment of the work.

REVIEWERS' COMMENTS:

Reviewer #1 (Remarks to the Author):

I thank the authors for considering and responding to my comments. The changes in the manuscript have addressed my concerns and I feel that the distinction between self-reported and actual walking pace is now made clearly and discussed sufficiently in the revised manuscript.

Reviewer #2 (Remarks to the Author):

The authors have been very responsive to the previous round of review. Reconciliation of data between the manuscript main text and Supplementary tables are now complete. I am particularly happy with the revised discussion section, as it critically appraises the contribution of BMI into the phenotype of self-reported walking. The authors note that although there is some correlation, most of the genetic association is independent.

CC Khor